# Genome-wide identification and expression profiling of the *COBRA-like* genes reveal likely roles in stem strength in rapeseed (*Brassica napus* L.)

Qian Yang, Shan Wang, Hao Chen, Liang You, Fangying Liu, Zhongsong Liu 🆔 *

College of Agronomy, Hunan Agricultural University, Changsha, Hunan, China

* zsliu48@sohu.com

**Data Availability Statement:** All relevant data are within the paper and its Supporting Information files.

**Funding:** This research was funded by the Scientific Research Foundation of the Hunan

## Abstract

The *COBRA-like* (*COBL*) genes play key roles in cell anisotropic expansion and the orientation of microfibrils. Mutations in these genes cause the brittle stem and induce pathogen responsive phenotypes in *Arabidopsis* and several crop plants. In this study, an *in silico* genome-wide analysis was performed to identify the *COBL* family members in *Brassica*. We identified 44, 20 and 23 *COBL* genes in *B. napus* and its diploid progenitor species *B. rapa* and *B. oleracea*, respectively. All the predicted COBL genes were phylogenetically clustered into two groups: the AtCOB group and the AtCOBL7 group. The conserved chromosome locations of *COBLs* in *Arabidopsis* and *Brassica*, together with clustering, indicated that the expansion of the COBL gene family in *B. napus* was primarily attributable to whole-genome triplication. Among the *BnaCOBLs*, 22 contained all the conserved motifs and derived from 9 of 12 subgroups. RNA-seq analysis was used to determine the tissue preferential expression patterns of various subgroups. *BnaCOBL9*, *BnaCOBL35* and *BnaCOBL41* were highly expressed in stem with high-breaking resistance, which implies these AtCOB subgroup members may be involved in stem development and stem breaking resistance of rapeseed. Our results of this study may help to elucidate the molecular properties of the *COBRA* gene family and provide informative clues for high stem-breaking resistance studies.

## Introduction

Plant morphogenesis is dependent on the regulation of cell division and expansion. Most plant cells grow anisotropically through internal and isotropic turgor pressure yield from cell walls [1]. The plant cell wall is a dynamic, complex fibrillar network. After the plant cell expands to its final shape and the primary cell wall is formed, the secondary cell wall is formed and thickens between the primary cell wall and plasma membrane [2, 3]. The *COBRA* gene, which encodes a glycosylphosphatidylinositol (GPI) anchored protein [1, 4, 5], regulates microfibril deposition on the cell surface at the rapid elongation stage to guarantee a normal anisotropic expansion of the cell wall during plant morphogenesis.

Provincial Education Department (grant number: 20A261), the National Key Research and Development Program of China (grant number: 2017YFD0101702) and the Key Research and Development Program of Hunan Province (grant number: 2016JC2024). The funders had no role in study design, data collection and analysis, decision to publish, or preparation of the manuscript.

**Competing interests:** The authors have declared that no competing interests exist.

The *COBRA* gene belongs to the COBRA-like (COBL) gene family. COBL proteins often contain an N-terminal secretion signal, a COBRA domain, potential N-glycosylation sites, a CCVS (Cys-rich) motif, and an *ω*-attachment site for GPI modification along with a hydrophobic C-terminal [1, 5]. Some of these proteins contain a predicted cellulose-binding site (CBM). The analysis of the *cob* allele indicated that *COBLs* can further affect the cellulose crystallinity status and cellulose content of the secondary cell wall [1].

The COBL family is conserved in monocots and eudicots [2, 5]. In *Arabidopsis* (*Arabidopsis thaliana*), there are 12 *COBLs* (*AtCOBLs*), which can be divided into two groups based on their protein sequences [1], one showing strong similarity to *COBRA* while the other exhibiting high similarity to *AtCOBL7*. There are 11, 11, and 10 *COBLs* that have been identified in the monocots rice (*Oryza sativa ssp. japonica*) [6], maize (*Zea mays*) [2], and sorghum (*Sorghum bicolor*) [7], respectively. Additionally, 17, 18, 24, and 33 *COBLs* have been reported in the eudicots tomato (*Solanum lycopersicum*) [8], *Populus* (*Populus* L.) [9], soybean (*Glycine max*) [10], and cotton (*Gossypium* spp.) [11], respectively. It appears that the family members increased to a certain extent in eudicots but remained almost constant in monocots. This phenomenon of expansion is presumed to be derived from whole-genome duplication [10, 11] and segmental duplication [5]. The phylogenetic relationship was similar to that of *Arabidopsis* among various reported species [10]. In the COBRA group, the AtCOB orthologous subgroup was predicted to be a sister clade of the AtCOBL4 subgroup and derived more recently after the division between monocots and eudicots [5].

The *COBLs* members have been found to mediate diverse physiological and developmental processes such as stem strength [6], pollen tube growth [12], pathogen resistance [13], and root-hair growth [4]. Silencing a *COBL* member, such as *BRITTLE CULM1 (OsBC1)* in rice, *Brittle stalk 2 (ZmBk2)* in maize, *BRITTLE CULM1 (SbBC1)* in sorghum, and *TmBr1* in diploid wheat, caused plants to exhibit the brittle phenotype [6, 7, 14, 15]. Cuticle lacking, abnormal shape, and irregular size distribution were observed in the epidermal cells of a tomato mutant in which the *SlCOBRA-like* gene was repressed. These phenotypes resulted in extensive non-uniform cracking on the surface of the immature green fruits of these plants [8]. Mutations in *AtCOBL10* were observed to cause gametophytic male sterility due to reduced pollen tube growth and compromised directional sensing in the female transmitting tract [12, 16, 17].

Rapeseed (*Brassica napus* L. AACC, $2n = 38$) which supplies approximately 13–16% of vegetable oil worldwide [18], is an allotetraploid species that was formed approximately 7,500–12,500 years ago by a spontaneous cross of the diploid progenitors *B. rapa* (AA, $2n = 20$) and *B. oleracea* (CC, $2n = 18$) [19]. In this study, we identified *COBL* genes at the genome-wide level and performed a comprehensive *in silico* analysis including characterization of phylogeny, gene structure, conserved motifs, and chromosomal collinearity in rapeseed and its progenitors. We also evaluated the expression patterns of these genes in various tissues as well as stems with different stem breaking resistance (SBR) by transcriptome sequencing. Our results may help to further characterize the functions of COBL family, and provide clues for stem strength in rapeseed.

## Materials and methods

### Genome-wide identification of *COBLs* in *Brassica napus* and its both progenitor species

The *B. napus* (*cv*. ZhongShuang11, ZS11) genome sequence was downloaded from the BnPIR database (http://cbi.hzau.edu.cn/bnapus/index.php) [18, 20]. The genome sequences, CDSs and annotation files of *B. rapa* (v3.0) and *B. oleracea* (HDEM) were retrieved from the *Brassica* Database (BRAD, http://brassicadb.cn). The *Arabidopsis* COBL protein sequences were

obtained from TAIR (http://www.arabidopsis.org) [5], and used as the query to identify COBL homologs in *B. napus*, *B. rapa* and *B. oleracea* by BLASTP [21], with the e-value being 1E-10. After redundant sequences and incomplete sequences were removed, the remaining protein sequences were submitted to SMART tools and the NCBI Conserved Domain Search Database to confirm the presence of previously characterized domains in the candidate sequences; sequences without COBRA domains were excluded from the downstream analysis [22].

The physicochemical parameters of BnaCOBL proteins, including the molecular weights (in kDa) and isoelectric points (pIs), were calculated by ExPASy [23]. The subcellular location of COBL proteins were predicted by Cell-PLoc v2.0 (http://www.csbio.sjtu.edu.cn/bioinf/Cell-PLoc-2/).

## Multiple alignments and phylogenetic analysis of *COBLs* from *Brassica* and *Arabidopsis*

All the predicted COBL protein sequences of *B. napus*, *B. rapa*, and *B. oleracea*, and the AtCOBLs protein sequences were aligned by Multiple Sequence Comparison by Log-Expectation (MUSCLE) [24]. The phylogenetic tree was generated in IQ-tree [25] software using the maximum likelihood (ML) method with 10,000 bootstrap replicates. The "figtree" (http://tree.bio.ed.ac.uk/software/figtree/) was used to draw the phylogenetic tree of COBL protein in four genomes.

## Chromosomal locations and syntenic analyses of *COBLs* in *Brassica napus* and its both progenitor species

The chromosomal positions of the *BnaCOBLs* were obtained from the genome annotation file of ZS11. The start and end locations of each *BnaCOBL* were drawn on chromosomes using MapChart [26]. The synteny relationships between the *BnaCOBLs* and *COBLs* in *B. rapa*, and *B. oleracea* were evaluated using the McScanX [27] and drawn by TBtools [28].

## Prediction of gene structures, conserved motifs, and *cis*-acting regulatory elements of *BnaCOBLs*

The gene structures (exon-intron) of *BnaCOBLs* were retrieved from the genome annotation file. The COBRA domain and potential N-glycosylation sites were predicted by GenomeNet Bioinformatics Tools (https://www.genome.jp/) and the NetNGlyc 1.0 server (http://www.cbs.dtu.dk/services/NetNGlyc/) [29]. The signal peptide, CCVS Cys-rich domain, and potential $\omega$-sites for GPI modification were predicted with Signal 5.0 [30] and the GPI Prediction Server Version 3.0 [31]. Hydrophilicity analysis was performed by ExPASy-ProtScale (https://web.expasy.org/protscale/) [32, 33]. TBtools was used to draw the structural map of *BnaCOBLs*. To further analyze the COBRA domains of *BnaCOBLs*, the multi-sequence alignments were carried out by MEGA v7.0 [34] and the results were displayed by GeneDoc (http://www.cris.com/~Ketchup/ genedoc.shtml).

To analyze the putative cis-regulatory elements (CAREs) of *BnaCOBLs*, the promoter regions were defined as the 1.5-kb region upstream of the ATG start codon of each gene (i.e., the 1.5-kb downstream sequences were chosen if a gene was found to map on the opposite strand relative to the sequence strand deposited in the ZS11 genome). These sequences were used to detect the CAREs with the online database PlantCARE [35]. Next, considering the characters of plant core promoter regions, we checked the common promoter elements TATA-box and CAAT-box near the start codon (<500bp), the core promoter elements (i.e., TATA-box, CAAT-box) on the opposite strands of the corresponding genes were filtered out

of the results because the core promoter regions are direction-sensitive [36]. We classified all the elements into core promoter elements, responsive elements, the temporal and spatial specific or unannotated elements according to their functional annotation.

## Expression analysis of *BnaCOBLs* in various tissues

The RNA-seq data obtained from 12 tissues of the rapeseed cultivar ZS11, which was described in a previous study [37], were downloaded from National Center for Biotechnology Information (NCBI) (ID: PRJNA394926) to assess the tissue expression preference of different COBL family members of rapeseed.

For further evaluation of the expression profiles of *BnaCOBLs* in rapeseed stem, we selected previously reported [38] transcriptome expression data of four stem samples: FH (High stem breaking resistance (SBR) during Flowering), FL (Low-SBR during Flowering), SH (High-SBR during Silique development), and SL (Low-SBR during Silique development). The high SBR sample had averaged SBR of 115.49N; while the low SBR sample had averaged SBR of 31.69N [38]. The raw data were downloaded from the Short Read Archive (SRA) database of NCBI under the accession number SRP142441.

The NGSQCToolkit [39] was used to clean the raw data. The RSEM [40] and STAR [41] softwares were used to map the clean reads to the reference genome of ZS11 and calculate the transcripts per million (TPM) values of each gene, and the heat map of expression of *BnaCOBL* genes was drawn by TBtools.

## Plant materials and qRT-PCR analysis

The seed of ZS11, a semi-winter rapeseed cultivar, was kindly provided by Oil Crops Research Institute, Chinese Academy of Agricultural Sciences and sown on the experimental farm of Hunan Agricultural University, Changsha. Three individual plants were harvested at the initial flowering stage. Their stems were cut into two parts, the upper (adjacent to inflorescence) and the lower (the first elongated internode). Fully expanded leaves were used as leaf samples whereas the taproot and the lateral roots were collected separately after being cleaned up.

Quantitative real-time RT–PCR (qRT–PCR) was performed to determine gene expression level. Total RNA was extracted from all sample tissues separately using an RNAqueous kit (Thermo Fisher, AM1912). The yield of RNA was determined using a NanoDrop 2000 spectrophotometer (Thermo Scientific, USA), and the integrity of the RNA was evaluated using agarose gel electrophoresis and staining with ethidium bromide. Each RT reaction consisted of 0.5 μg RNA, 2 μl of 5X TransScript All-in-One SuperMix for qPCR and 0.5 μl of gDNA Remover in a total volume of 10 μl. Reactions were performed in a GeneAmp® PCR System 9700 (Applied Biosystems, USA) for 15 min at 42˚C and 5 s at 85˚C. The 10-μl RT reaction mix was subsequently diluted tenfold in nuclease-free water. Real-time PCR was performed using LightCycler® 480 II Real-time PCR Instrument (Roche, Swiss) with 10 μl PCR reaction mixture that included 1 μl of cDNA, 5 μl of 2X PerfectStart™ Green qPCR SuperMix, 0.2 μl of forward primer, 0.2 μl of reverse primer and 3.6 μl of nuclease-free water. Reactions were incubated in a 384-well optical plate (Roche, Swiss) at 94˚C for 30 s followed by 45 cycles of 94˚C for 5 s and 60˚C for 30 s. Each sample was repeated three times. The expression levels of mRNAs were normalized to *BnaActin* and were calculated using the comparative cycle threshold (Ct) method [42]. The primers were designed at the specific nucleotide among the CDSs of five *BnaCOBLs* and checked through electronic PCR on the CDSs of these genes. These primer sequences are listed in the S1 Table.

## Results

### Identification of the *COBL* genes in *Brassica napus* and its diploid progenitors

A total of 62 putative *COBLs* were identified in *B. napus* through a BLASTP search using 12 *Arabidopsis* COBL protein sequences as query. These sequences were submitted to SMART and the NCBI CDD (Conserved Domains Database) to confirm the existence of COBRA domains. Finally, 44 candidate *COBLs* were identified and designated as *BnaCOBL1-44* in rapeseed, and their basic information is listed in the S2 Table. Among these proteins, *BnaCOBL38* was determined to be the largest with 699 amino acids (aa), whereas *BnaCOBL19* was the smallest with 200 aa. The molecular weights and isoelectric points of the *BnaCOBLs* ranged from 22.06 to 77.68 kDa and 5.25 to 10.09 (S2 Table), respectively. The *BnaCOBLs* were predicted to localize at the cell membrane (30), extracellular (12), and endoplasmic reticulum (2).

Similarly, we also identified 20 *BraCOBLs* and 23 *BolCOBLs* in *B. rapa* and *B. oleracea*, both progenitor species of *B. napus*, respectively. Their gene symbols and chromosomal locations are listed in S2 Table. There were approximately two times as many *COBLs* in *B. rapa* and *B. oleracea* as in *Arabidopsis*. The sum of *COBLs* in the diploid progenitors was almost equal to the quantity of *BnaCOBLs*.

### Phylogenetic analysis of the *COBL* genes from *B. napus, B. rapa, B. oleracea* and *Arabidopsis*

To unravel the evolutionary relationships among the *COBL* genes from *B. napus, B. rapa, B. oleracea* and *Arabidopsis*, a phylogenetic tree was constructed based on whole protein sequences using ML method. As shown in Fig 1, all the *COBL* members were clustered into two groups, which corresponded with the AtCOB group and the AtCOBL7 group in *Arabidopsis* [5]. The AtCOB group (Group I) contained *AtCOB, AtCOBL1-6*, 12 *BraCOBLs*, 15 *BolCOBLs*, and 25 *BnaCOBLs*, while the AtCOBL7 (Group II) consisted of *AtCOBL7-11*, 8 *BraCOBLs*, 8 *BolCOBLs*, and 19 *BnaCOBLs*. Group I contained more *COBLs* than Group II in the four species analyzed.

Based on the bootstrap values and the topology of the phylogenetic tree, these proteins were further divided into 12 subgroups (Table 1). Each subgroup had *COBLs* from four species, except the BnaCOBL5/44 subgroup, which lacks *COBLs* from *Arabidopsis*. The subgroups AtCOBL1, AtCOBL7, AtCOBL8, and AtCOBL9 each retained two *BnaCOBLs*, while six and eight *BnaCOBLs* were retained in the subgroups AtCOBL2/3 and AtCOBL11, respectively. The other subgroups had three or five BnaCOBLs. These results indicated an unequal evolution among orthologous subgroups of *BnaCOBLs* when derived from corresponding *AtCOBLs*. The subgroup AtCOB was closer to AtCOBL5 in Group I. while the subgroup AtCOBL10 was closer to AtCOBL11 in Group II. This distribution was similar to that in *Arabidopsis*. Based on the triploidy and allotetraploidization events in the evolutionary history of rapeseed, each subgroup of this phylogenetic topology represented a class of orthologous *COBLs* in *Brassica* species derived from the corresponding *AtCOBL*.

### Chromosomal locations of *COBLs* and syntenic analyses between *Brassica napus* and its progenitor

The *BnaCOBLs* were unevenly distributed on 16 of 19 chromosomes (except for A04, C04 and C06) of rapeseed, with one to five members on each chromosome (Fig 2 and S2 Table). The *BnaCOBLs* were asymmetrically distributed in subgenomes: 19 were detected in the A subgenome, and 25 were detected in the C subgenome. However, the locations of *BnaCOBLs* on

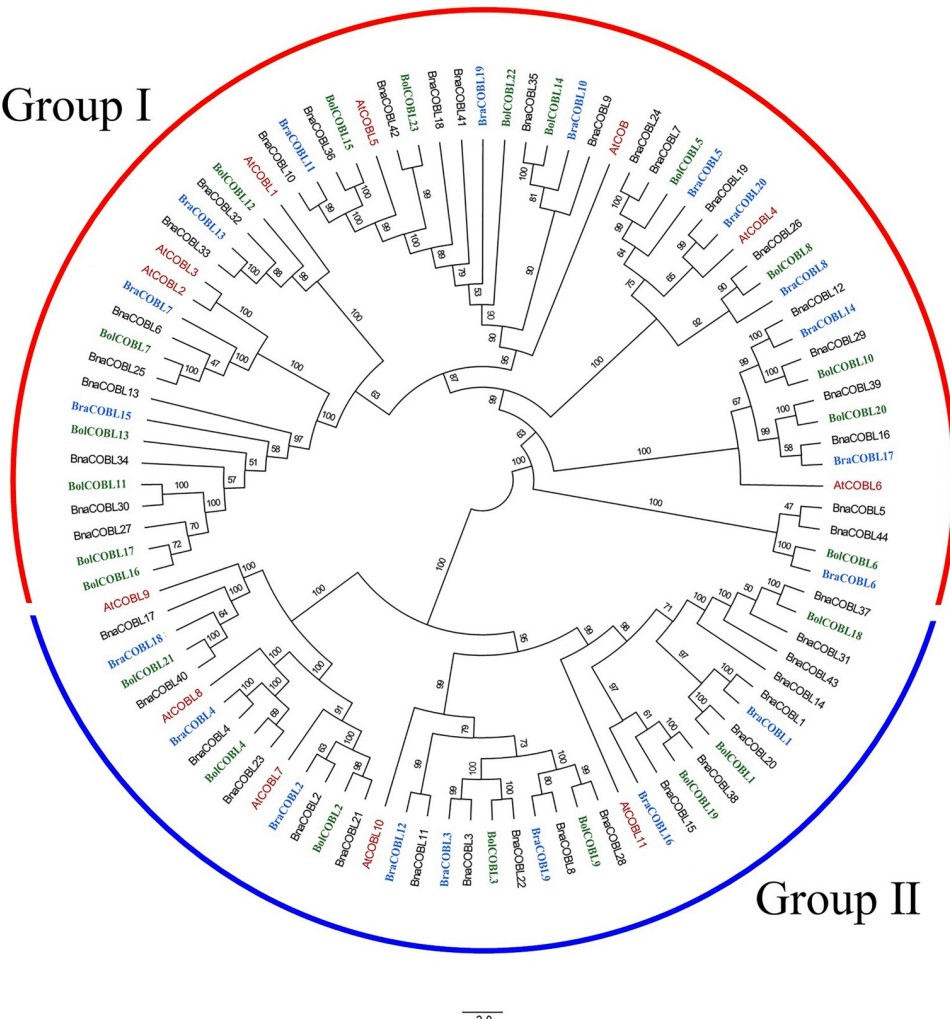

**Fig 1. Phylogenetic analysis of COBL proteins in *Brassica napus*, *B. rapa*, *B. oleracea*, and *Arabidopsis*.** All protein sequences were aligned by MUSCLE software. The phylogenetic tree was constructed by IQ-tree by ML method with 10,000 bootstrap replicates. These proteins were clustered into two groups. The red, blue, and green font represent the *COBLs* of *Arabidopsis*, *B. rapa*, and *B. oleracea*, respectively. The percentages of bootstrap numbers for the nodes are displayed on the branches.

chromosome A01, A02, and A09 were much the same as the locations of their homologous C01, C02, and C09. Even the *BnaCOBLs* on A01 and *BnaCOBLs* at the homologous region on C01 were determined to belong to the same subgroups. This kind of gene pairs were also observed on some other homologous chromosomes.

Chalhoub et al [19] reported that 80.0% of genes in *B. napus* (*cv*. Damor) were orthologous to the genes of *B. rapa* and *B. oleracea*. Based on protein sequence identity and phylogenetic topology, we identified 18 and 21 orthologous gene pairs (S3 Table) between the subgenomes of rapeseed and their respective ancestral genomes. The locations of these orthologous pairs of *COBLs* showed high similarity between *B. napus* and *B. rapa* or *B. oleracea*, respectively (Fig 3A). We found two *BraCOBLs* (*BraCOBL5* and *BraCOBL13*) and three *BolCOBLs* (*BolCOBL6*, *BolCOBL16*, and *BolCOBL17*) have lost their orthologous gene pair in *B. napus*. On the other hand, the five *BnaCOBLs* (*BnaCOBL14*, *BnaCOBL27*, *BnaCOBL31*, *BnaCOBL43*, and *Bna-COBL44*) did not detect orthologs in either *B. rapa* or *B. oleracea*.

**Table 1. Domains of BnaCOBL proteins in rapeseed.**

| Gene Name | Chr. | Subgroup | CCVS[1] | N-terminal secretion signal cleavage site | ω-site[2] Position | p-value | Hydrophobic C-terminal |
|---|---|---|---|---|---|---|---|
| *BnaCOBL9* | A03 | AtCOB | 233 | TEA-YD | N431 | 2.3*e-07 | yes |
| *BnaCOBL35* | C07 | AtCOB | 233 | TEA-YD | N431 | 2.4*e-06 | yes |
| *BnaCOBL41* | C09 | AtCOB | 233 | TEA-YD | N431 | 2.2*e-07 | yes |
| *BnaCOBL32* | C05 | AtCOBL1 | 231 | ADA-YD | N428 | 3.7*e-04 | yes |
| *BnaCOBL33* | C05 | AtCOBL1 | 232 | ADA-YD | *A428* | 2.9*e-04 | yes |
| *BnaCOBL27* | C03 | AtCOBL2/3 | | - | *A187* | - | no |
| *BnaCOBL6* | A02 | AtCOBL2/3 | 223 | TEA-YD | N415 | 2.8*e-05 | yes |
| *BnaCOBL13* | A06 | AtCOBL2/3 | 223 | TEA-YD | N412 | 2.5*e-07 | yes |
| *BnaCOBL25* | C02 | AtCOBL2/3 | 223 | TEA-YD | N415 | 2.1*e-04 | yes |
| *BnaCOBL30* | C05 | AtCOBL2/3 | | - | *G296* | - | no |
| *BnaCOBL34* | C07 | AtCOBL2/3 | 223 | TEA-YD | N412 | 1.6*e-07 | yes |
| *BnaCOBL7* | A03 | AtCOBL4 | | ASA-YD | *W526* | - | yes |
| *BnaCOBL19* | A10 | AtCOBL4 | | TSA-YD | *G170* | - | no |
| *BnaCOBL24* | C02 | AtCOBL4 | | SSA-YD | *G215* | - | no |
| *BnaCOBL26* | C03 | AtCOBL4 | | ASA-YD | *M204* | - | no |
| *BnaCOBL10* | A03 | AtCOBL5 | | SEA-LT | *M184* | - | no |
| *BnaCOBL18* | A09 | AtCOBL5 | | TEA-YD | *G382* | - | no |
| *BnaCOBL36* | C07 | AtCOBL5 | | SEA-LT | *S209* | - | no |
| *BnaCOBL42* | C09 | AtCOBL5 | | - | *T183* | - | no |
| *BnaCOBL12* | A06 | AtCOBL6 | 222 | SHG-YD | *S270* | - | no |
| *BnaCOBL16* | A08 | AtCOBL6 | 219 | THG-FD | S412 | 1.9*e-07 | yes |
| *BnaCOBL29* | C05 | AtCOBL6 | 221 | SHG-YD | *R548* | - | no |
| *BnaCOBL39* | C08 | AtCOBL6 | 218 | THG-FD | S411 | 1.1*e-07 | yes |
| *BnaCOBL5* | A02 | - | | SLG-RY | *M186* | - | no |
| *BnaCOBL44* | C09 | - | | - | *M499* | - | no |
| *BnaCOBL2* | A01 | AtCOBL7 | 420 | TTS-QS | S635 | 2.3*e-05 | yes |
| *BnaCOBL21* | C01 | AtCOBL7 | 419 | TAS-QS | N635 | 3.0*e-05 | yes |
| *BnaCOBL4* | A01 | AtCOBL8 | 427 | TSS-QP | S642 | 6.6*e-06 | yes |
| *BnaCOBL23* | C01 | AtCOBL8 | 423 | TSS-QQ | N638 | 9.3*e-06 | yes |
| *BnaCOBL17* | A09 | AtCOBL9 | 421 | SLS-QL | G638 | 9.8*e-05 | yes |
| *BnaCOBL40* | C09 | AtCOBL9 | 421 | SLS-QL | S638 | 2.3*e-05 | yes |
| *BnaCOBL3* | A01 | AtCOBL10 | 433 | CNG-QD | S646 | 1.6*e-05 | yes |
| *BnaCOBL8* | A03 | AtCOBL10 | 432 | CNG-QD | S645 | 1.3*e-05 | yes |
| *BnaCOBL11* | A05 | AtCOBL10 | | - | *G373* | - | no |
| *BnaCOBL22* | C01 | AtCOBL10 | 433 | CNG-QD | *S646* | - | yes |
| *BnaCOBL28* | C03 | AtCOBL10 | 422 | CNG-QD | S635 | 9.2*e-06 | yes |
| *BnaCOBL1* | A01 | AtCOBL11 | 423 | SFA-QD | S635 | 2.4*e-05 | yes |
| *BnaCOBL14* | A07 | AtCOBL11 | | - | *A237* | - | no |
| *BnaCOBL15* | A08 | AtCOBL11 | 428 | SLA-QD | *Y641* | - | yes |
| *BnaCOBL20* | C01 | AtCOBL11 | 425 | SRA-QD | S637 | 3.0*e-05 | yes |
| *BnaCOBL31* | C05 | AtCOBL11 | | - | *S198* | | yes |
| *BnaCOBL37* | C08 | AtCOBL11 | | - | *E188* | - | no |
| *BnaCOBL38* | C08 | AtCOBL11 | 458 | - | *S670* | - | yes |
| *BnaCOBL43* | C09 | AtCOBL11 | | - | *S198* | | yes |

[1]The start site of the CCVS domain.

[2]The amino acid and location of the ω-site (GPI attachment cleavage site). A site represented in italics means that the confidence of this prediction did not reach the threshold.

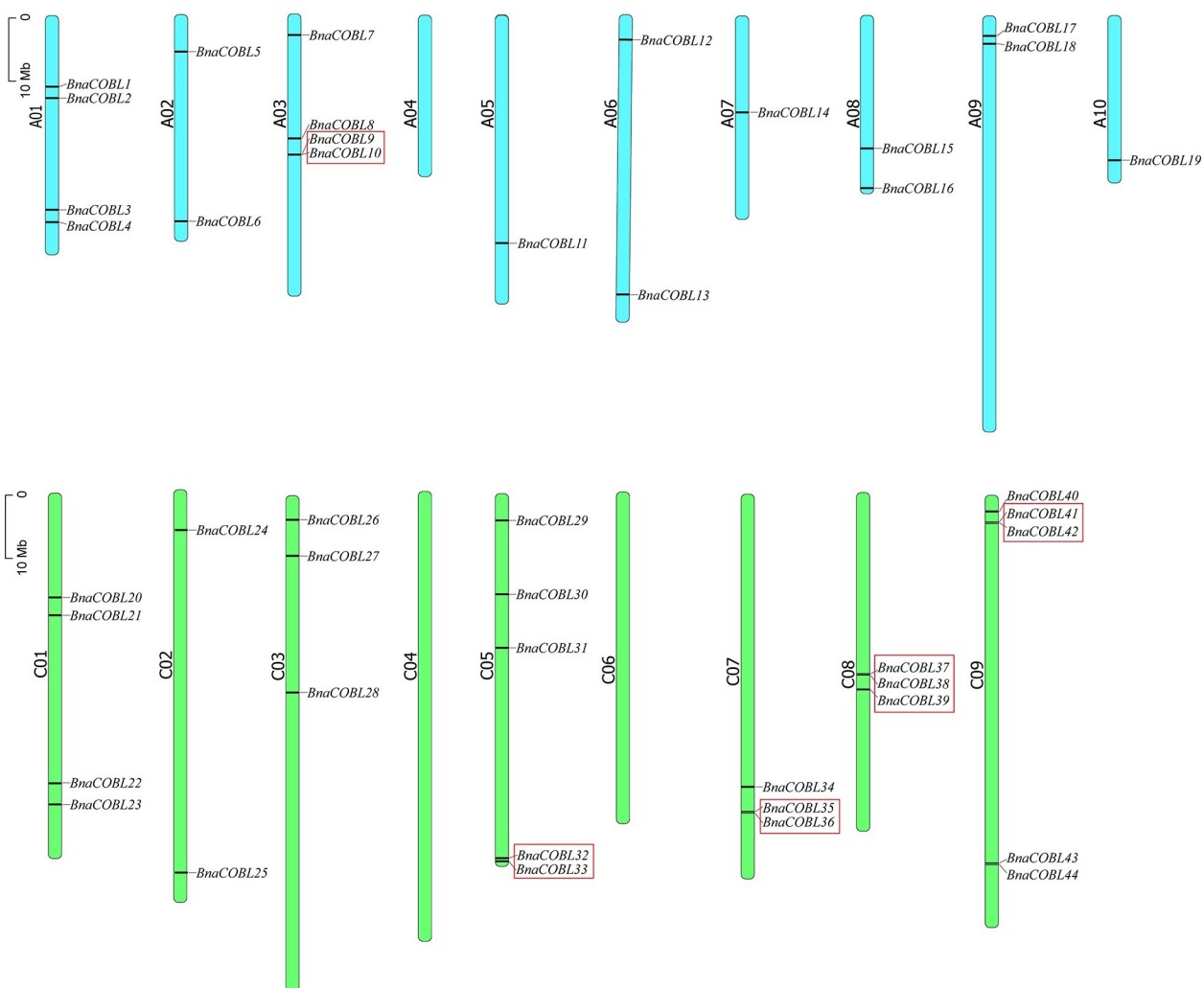

**Fig 2. Chromosomal locations of *BnaCOBLs*.** The blue and the green bar represent A- and C-subgenome chromosomes, respectively. The gene name is presented to the right of each bar, while the chromosome name is to the left. To the left of the A/C subgenome is a 10-Mb bar. The clusters are shown with a red frame.

The *BnaCOBL* gene clusters [43–45] containing two or three *BnaCOBLs* appear on the chromosome A03, C05, C07, C08, and C09 (Fig 2). The cluster on A03, C07, and C09 exhibited the same order on the "X block" [46] of *B. rapa*, *B. oleracea* and *Arabidopsis* (Fig 3B). The cluster on C08 was detected in *B. oleracea* but not in *Arabidopsis*. The cluster on C05 was detected only in *B. napus*, and this cluster may have been formed by segmental duplication in *B. napus* according to sequence and annotation of the ZS11 genome. These results suggested that the COBL gene family in rapeseed changed little during the allotetraploidization event from *B. rapa* and *B. oleracea*.

## Structure and conserved domains of *BnaCOBLs*

We characterized gene structure and motif domains of the *BnaCOBL* genes. These genes had 2–12 exons (Fig 4). The number of exons varies between two groups. In Group I, 17 of 25 members possessed over 6 exons, whereas all members in Group II were determined to have only two to four exons. However, the average length of proteins was observed to be longer in Group II than in Group I.

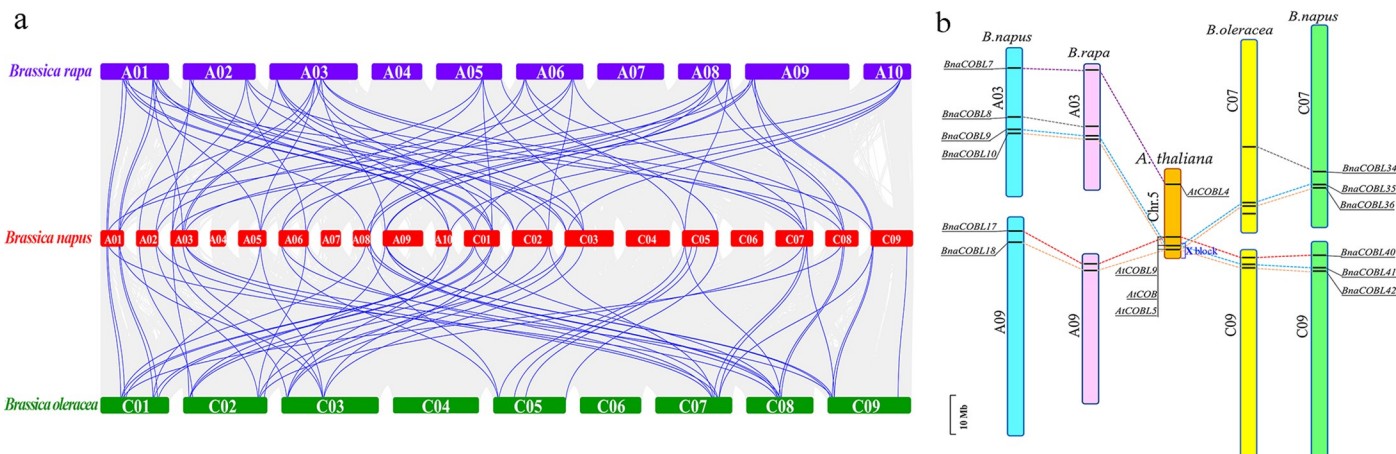

**Fig 3. Synteny of *COBLs* between *B. napus* and *B. rapa* or *B. oleracea*.** (a) Genome-wide synteny analysis for *COBLs* between *B. napus* and *B. rapa* or *B. oleracea*. The bars in red, purple, and green bars represent the chromosomes of *B. napus*, *B. rapa*, and *B. oleracea*, respectively. The homologous pairs of *COBLs* between *B. napus* and *B. rapa* or *B. oleracea* were connected with blue lines, (b) The syntenic relationships of gene clusters among *B. napus*, *B. rapa*, *B. oleracea*, and *Arabidopsis*. The blue square bracket marked the X block of *Arabidopsis*. The homologous *COBLs* of different genomes are connected by lines of the same color. A 10-Mb ruler is located in the lower-left corner.

All the *BnaCOBLs* were observed to have the COBRA domain (Fig 4). The COBRA domain of the Group I members is close to the N-terminal secretion signal peptide, while that of the Group II members is in the middle of the protein sequences. But there are exceptions to this rule. The multiple-sequence alinements showed eight *BnaCOBLs* whose COBRA domain is largely defective (S1 Fig). For example, *BnaCOBL31*, *BnaCOBL43* and *BnaCOBL37* only retained 29 to 58 amino acids at C-terminal. The COBRA domain showed significant divergence between 12 subgroups and high conservation within the subgroup although it had almost the same in one-third amino acid residues among family members. One or more potential N-glycosylation sites were distributed to all *BnaCOBLs*. Thirty-four of these proteins were identified beginning with an N-terminal secretion signal. The CCVS (Cys-rich) motif of 27 *BnaCOBLs* was observed to have seven to ten amino acids away from the C-terminal of the COBRA domain. Half of the family members (Fig 4 and Table 1) were determined to have all the conserved domains, so did the $\omega$-sites follow by a hydrophobic C-terminal domain.

Compared to *AtCOBL4*, *BC1*, *BK2*, the orthologous *BnaCOBLs* lost the C-terminal motif of COBRA domain, CCVS, and $\omega$-site (S2 Fig) because of the exon skipping (*BnaCOBL7* and *BnaCOBL26*) or the intron-retention (*BnaCOBL19* and *BnaCOBL24*). These similar alternative splicing events were also identified in other rapeseed sequenced genomes so that no complete *BnaCOBL7* can be found in the rapeseed pan-genome. The three members of AtCOB subgroup were conserved among all nine rapeseed genomes, which is shown in S2 Fig. In this subgroup, only two orthologous *COBLs* of *B. rapa* and *B. oleracea* were defective and also disappeared from *B. napus*.

## *Cis*-acting regulatory elements in the promoter region of *BnaCOBLs*

The control over gene transcription via upstream *cis*-acting regulatory elements (CAREs) is the most prominent mechanism governing gene expression regulation [47]. The analysis of CAREs may help elucidate the expression levels of *BnaCOBLs* in specific tissues and conditions [48]. To predict putative *cis*-elements in the *BnaCOBLs*, DNA sequences 1500 bp upstream of the start codon (ATG) were searched for in the PlantCARE database to identify the CAREs associated with plant growth, development, and stress response. Eighty-five CAREs were

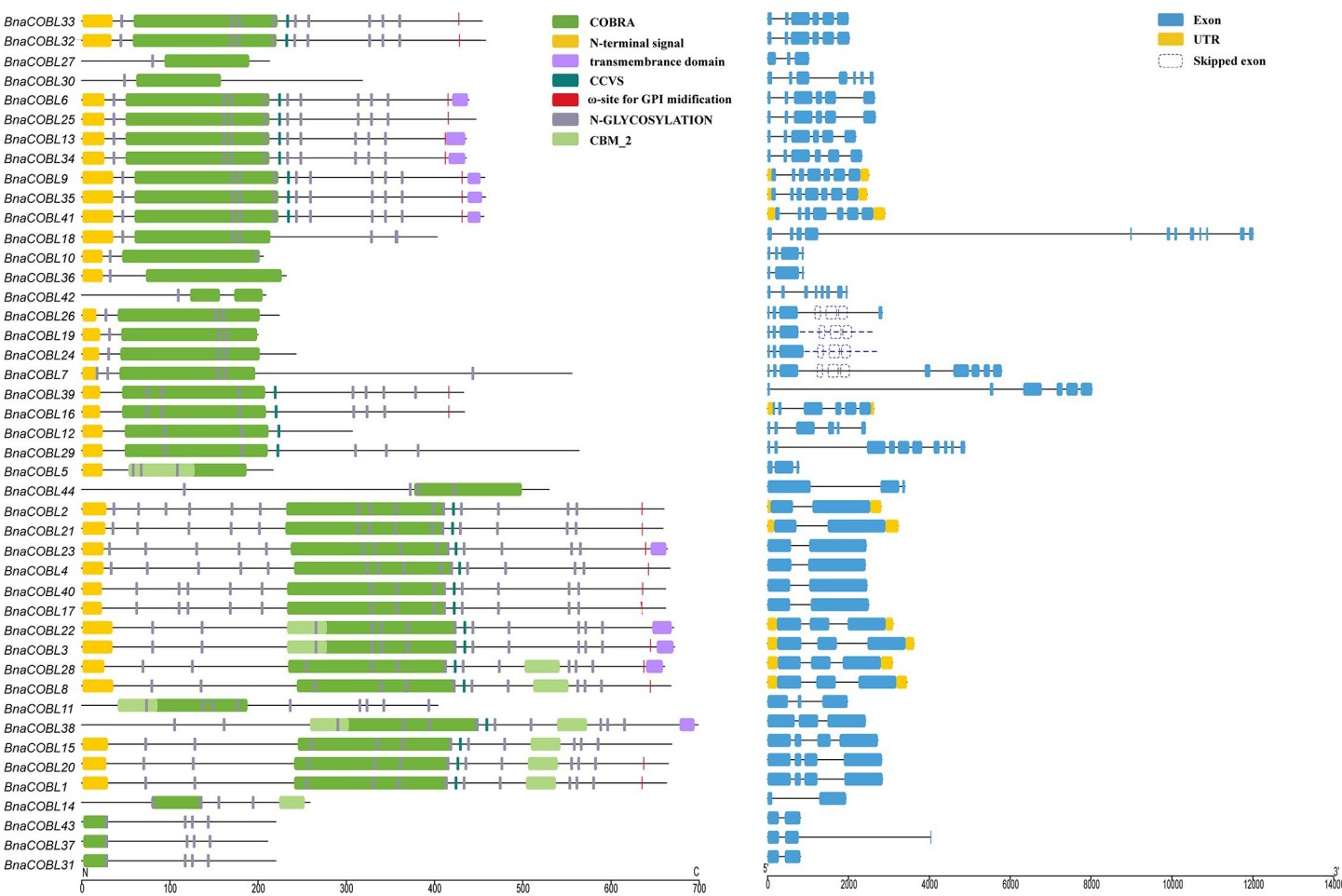

**Fig 4. Domain compositions and the gene structure of *BnaCOBLs*.** The dotted rectangles represent the skipped and missed exons according to how the orthologous genes are spliced in *B. rapa*, *B. oleracea*, and *Arabidopsis*. The dashed lines modify the genomic sequences.

found in all *BnaCOBLs*. All promoter regions of *BnaCOBLs* contained CAAT-box which is the major determinant of promoter efficiency. Three or more TATA-boxes were found in all the genes except two, *BnaCOBL10* and *BnaCOBL36* which had the TATA-less type of promoters [49].

We also analyzed the phytohormones and environment responsive elements (Fig 5 and S4 Table). The stress-related CAREs (S4 Table) were the most common and identified among all the *BnaCOBLs*. These stress-related CAREs included MYB, MYC, ARE and as-1, which correspond to abiotic and biotic stress. The most frequent stress CAREs were the MYCs, a dehydration-responsive element. The *BnaCOBLs* were probably regulated by methyl jasmonate (MeJA), ethylene (ETH), and abscisic acid (ABA) since these phytohormones-responsive CAREs were detected frequently in putative promoter regions. We found 22 light-responsive elements located in all *BnaCOBLs*. These discoveries indicated that *BnaCOBLs* could be regulated by stress-related, phytohormone-responsive, and light-induced transcription factors.

## Tissue specificity of expression of *BnaCOBLs* in rapeseed

The function of *COBLs* has been reported in root, flower, stem, and fruit skin of *Arabidopsis*, rice, maize, and tomato. We collected the RNA-seq data from the Sequence Read Archive

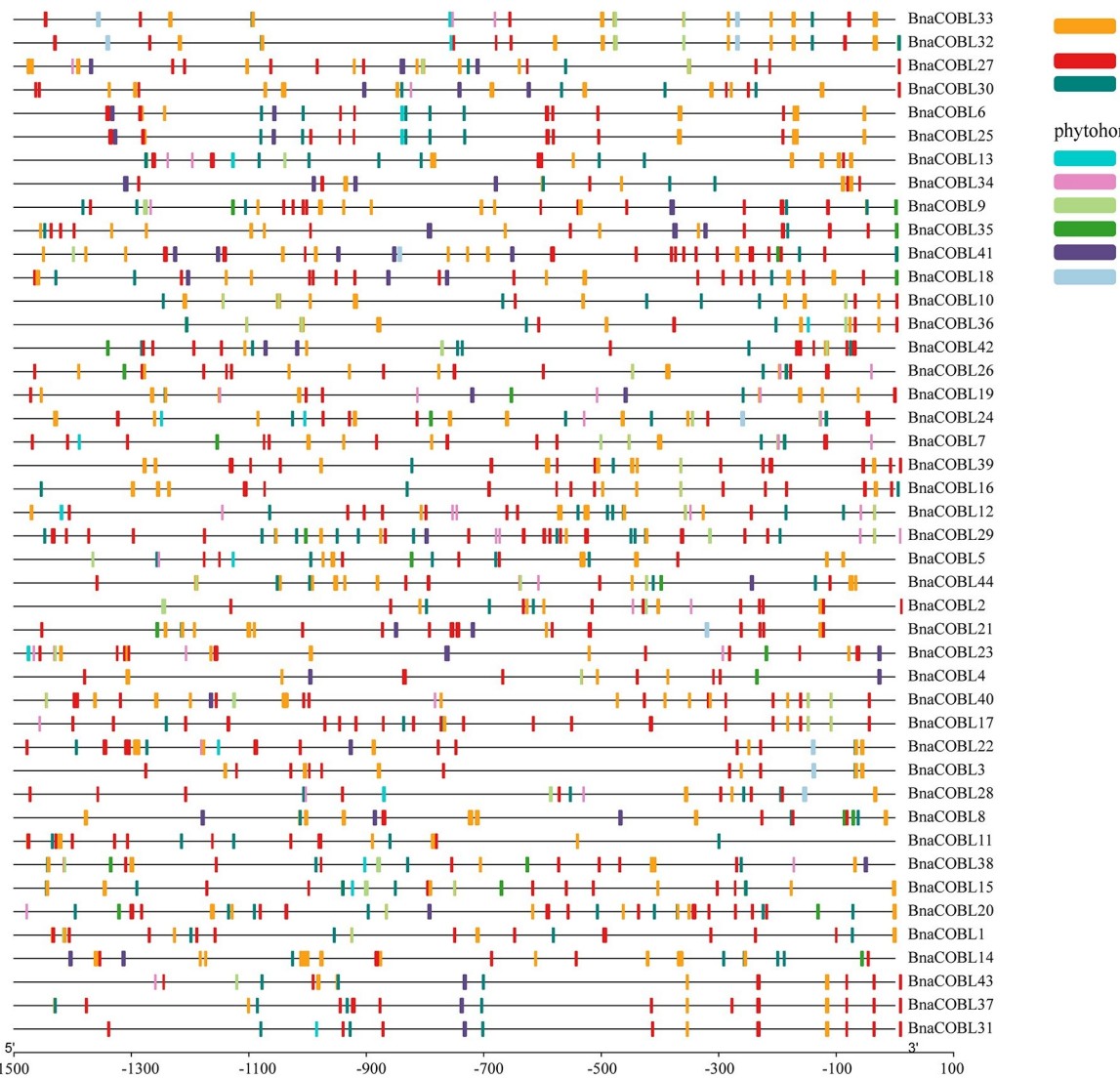

**Fig 5. *Cis*-acting regulatory elements in the promoter region of *BnaCOBLs*.** Different colors represent cis-regulatory elements with different predicted functions. In this figure, MYB represents MYB, MBS, MBSI, the MYB-like sequence, the Myb-binding site, and the MYB recognition site; ABA responsiveness represents ABRE, ABRE3a, and ABRE4; MeJA responsiveness represents the TGACG-motif and the CGTCA-motif; GA responsiveness represents the P-box, the GARE-motif, and the ATC-box; Auxin represents the TGA element and the AuxRR core, and stress responsiveness and light responsiveness covered 11 and 22 *cis*-regulatory elements respectively.

(SRA) to examine the tissue-specific expression of the 44 *BnaCOBLs*. These tissues include stem, leaf, root, flower, stamen, ovule, pistil, silique, sepal, pericarp, blossomy pistil, and wilting pistil. The TPM values are listed in the S5 Table.

The expression profiling (Fig 6) in the various tissues demonstrated that *BnaCOBLs* participated in biological processes in all examined tissues, especially the *COBLs* of the subgroups the AtCOB, AtCOBL7, and AtCOBL8, whereas the *COBLs* of the subgroups AtCOBL1, AtCOBL10, and AtCOBL11 were specifically expressed in floral organs. Even individual genes, such as *BnaCOBL6* and *BnaCOBL25*, were expressed in the ovule. The expression levels in tissues were mostly conserved in the intra orthologous subgroups but were different in inter orthologous subgroups. The eight members (*BnaCOBL5*, *BnaCOBL11*, *BnaCOBL14 BnaCOBL31*, *BnaCOBL37*, *BnaCOBL42*, *BnaCOBL43*, and *BnaCOBL44*) were found not to be

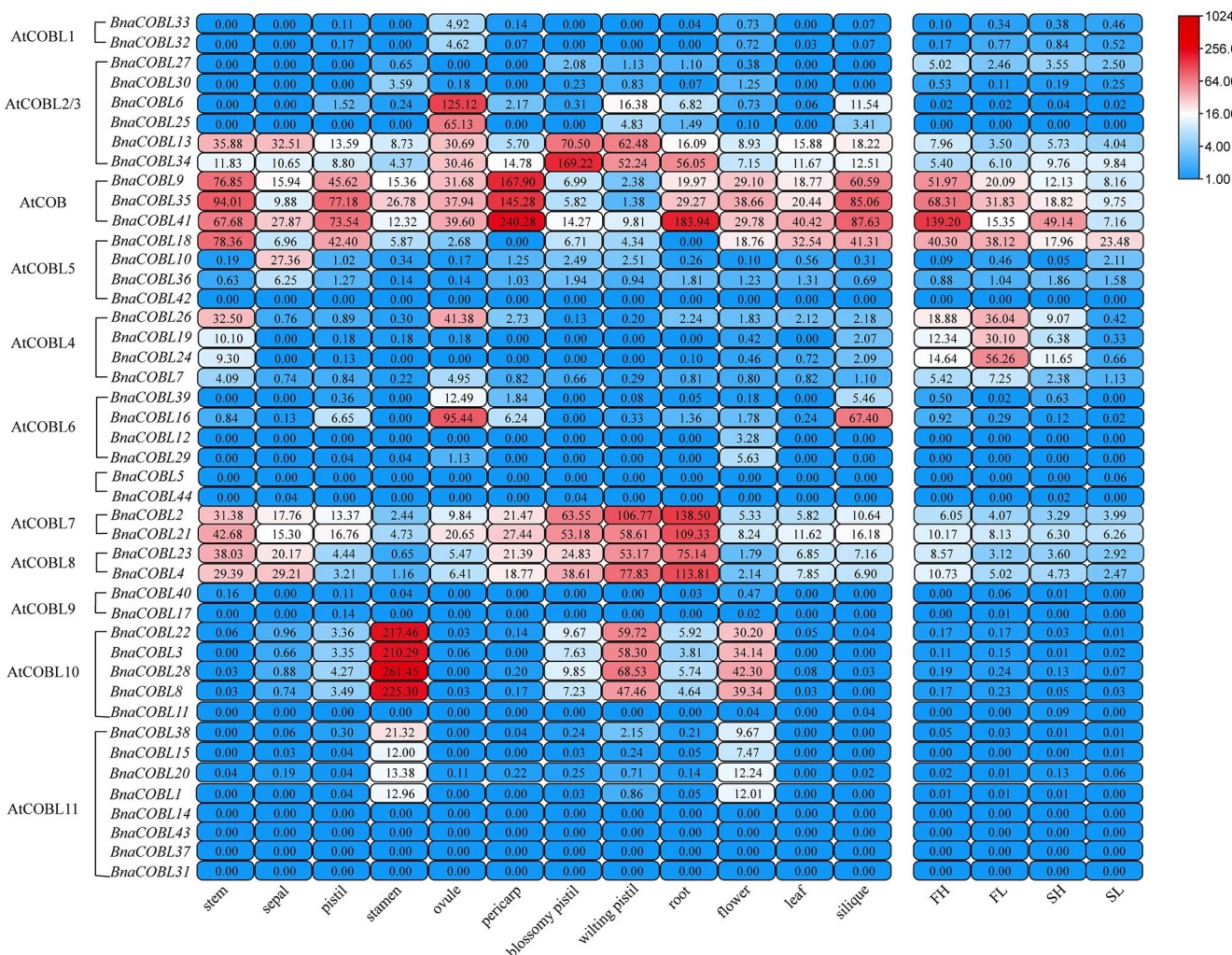

**Fig 6. Heatmap of the expression levels of *BnaCOBLs*.** Blue represents little or no expression, and red represents a high level of expression. The expression patterns of *BnaCOBLs* in various tissues and in the stem with different stem breaking resistance (SBR) are presented. The subgroups are noted to the left of the gene names.

expressed in any tissues. These expression characteristics of *BnaCOBLs* implied that specific subgroup members have different functions in different organs.

Considering the reported brittle culm mutants and the importance of stem stress resistance, we further compared the expression levels of *BnaCOBLs* in stems with different SBR levels. All three AtCOB subgroup members *BnaCOBL9*, *BnaCOBL35*, and *BnaCOBL41* were most active in the stem compared with other subgroups (Fig 6) and expressed at higher levels in the High-SBR one. Among these three genes, *BnaCOBL41* was expressed highest. In contrast, the AtCOBL4 subgroup members *BnaCOBL7*, *BnaCOBL26*, *BnaCOBL19*, and *BnaCOBL24*, which were found to be involved in stem breaking in cereal crops, were weakly expressed in the High-SBR stem.

To confirm these results, we selected the above three AtCOB subgroup members *Bna-COBL9*, *BnaCOBL35*, and *BnaCOBL41*, and two AtCOBL4 subgroup genes *BnaCOBL19*, and *BnaCOBL24* to quantify their expression with qRT-PCR in taproots, lateral roots, flower buds, leaves, upper and lower stems of rapeseed (ZS11) at the flowering stage. The amplification

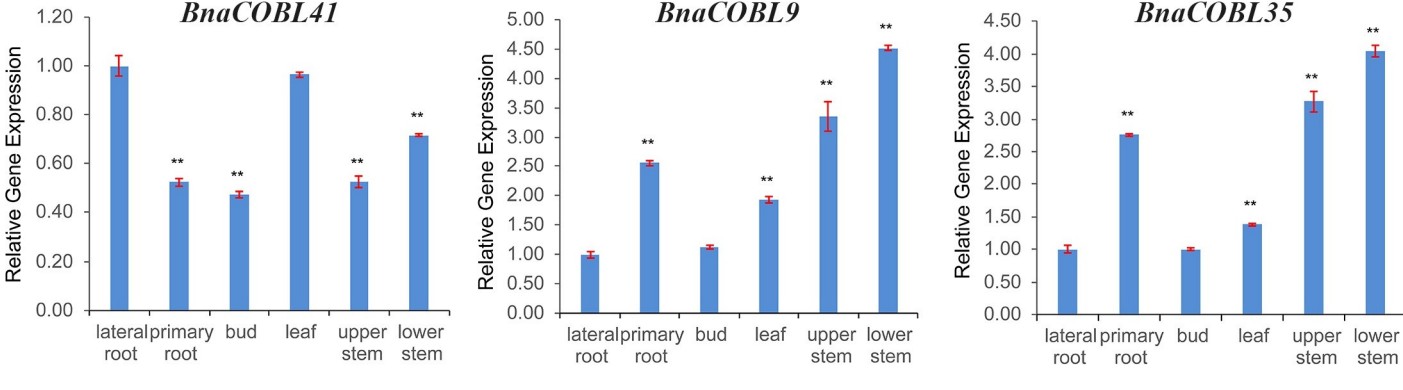

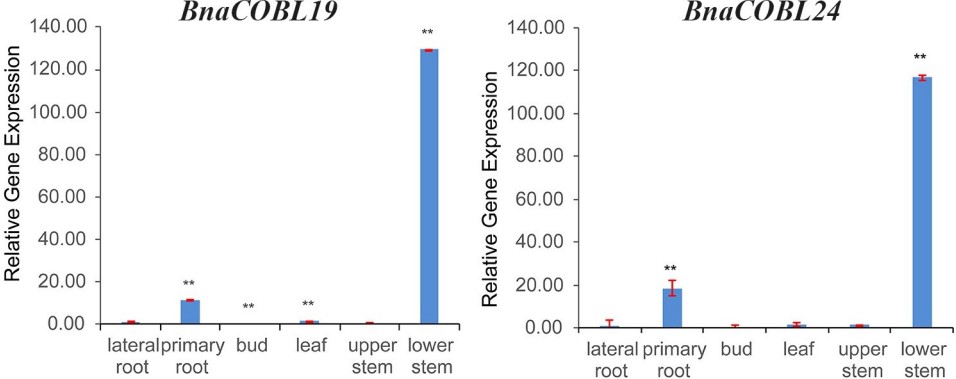

**Fig 7. qRT-PCR analyses of five selected *BnaCOBLs*.** A relatively high expression level was observed for the three *AtCOB* subgroup genes in the stem, root, and leaf, with the *BnaCOBL41* gene being expressed more strongly in the root, while the other two genes were expressed at higher levels in stems. The *AtCOBL4* subgroup genes were expressed at higher levels than other genes in these tissues. The stem adjacent to an inflorescence is defined as the "upper stem"; the first elongated node is defined as the "lower stem". Single and double asterisks represent differences from the control sample at the 5% and 1% significance levels, respectively. Error bars represent the standard deviations of three independent measurements.

curves showed that the average expression level of the AtCOB subgroup genes was higher in the stem than that of the AtCOBL4 members (Fig 7).

Furthermore, the results of qRT-PCR on various parts of stems implied that the genes *Bna-COBL9*, *BnaCOBL35*, and *BnaCOBL41* have different functions not only at different developmental stages but also in different internodes.

## Discussion

In this study, we identified 44 COBRA-like genes in rapeseed and analyzed their phylogenetic relationships, chromosome locations, domain composition, and putative *cis*-elements. Together with the tissue specific expression patterns, these characters were differentiated by subgroups which were orthologs from different *COBLs* of *Arabidopsis*.

The COBRA family has been reported in many species in the plant kingdom, even in the moss *Physcomitrella patens* [5]. This family has already emerged in the ancestor of *Arabidopsis*. In *Arabidopsis*, there are 12 *AtCOBLs*, and segmental duplication contributed to *AtCOBLs*, as two pairs of duplication had been identified (*AtCOBL2* and *AtCOBL3*, *AtCOBL1* and *AtCOBL4*) [5].

We identified 44 *BnaCOBLs*, 20 *BraCOBLs*, and 23 *BolCOBLs* in the genomes of the allotetraploid *Brassica napus* and its diploid progenitor species *B. rapa* and *B. oleracea*, respectively

(Fig 1). *Brassica* evolved from a Brassiceae lineage-specific whole genome triplication (WGT) [19] after diverged from a common ancestor with Arabidopsis about 20 million years ago [50, 51]. After WGT the number of *BraCOBLs* and *BolCOBLs* almost doubled compared to the number of *AtCOBLs*. However, the number of *BnaCOBLs* was close to the sum of the number of *BraCOBLs* and *BolCOBLs* after allopolyploidization. Furthermore, *BnaCOBLs* are highly syntenic to [52–55] and conserved in gene clusters of *BraCOBLs* and *BolCOBLs* (Fig 1). We propose that whole-genome triplication event contributed to the expansion of *BnaCOBLs*.

The expression profiling demonstrated expression patterns of *BnaCOBLs* in twelve tissues (Fig 6). As the stem with reinforced mechanical strength showed higher resistance to lodging and pathogen attack [13, 38]. We concentrated on their expression levels in stems with different breaking resistance and found all three *AtCOB* subgroup members *BnaCOBL9*, *Bna-COBL35*, and *BnaCOBL41* were expressed at higher levels in the High-SBR stem than in the Low-SBR one. *BnaCOBL9* is located near the lodging coefficient QTL on A03, and *Bna-COBL41* is located 300 kb upstream of the breaking force QTL on C09 in rapeseed [56]. Both *BnaCOBL9* and *BnaCOBL35* are reported to be hub genes with some *CesA* in a co-expression module, which was predicted to be relevant to cellulose biosynthesis [38]. We postulate that the AtCOB subgroup *BnaCOBLs* may play a role in the formation of stem strength in rapeseed.

Contrary to the expectation, the cloned AtCOBL4 subgroup *COBL* genes such as *BC1* in rice, *BK2* in maize, which were shown to be associated with the stem-breaking resistance in the grass family [57], were weakly expressed in High-SBR stems of rapeseed, which indicates AtCOBL4 subgroup members are not involved in the formation of stem strength in rapeseed. None of all AtCOBL4 subgroup members maintained all core motifs of *COBLs* (Fig 4), whereas members in this subgroup of *B. rapa* and *B. oleracea* were complete (S2 Fig). This structural change brought about alternative splicing variants, that is, exon skipping (*BnaCOBL7* and *Bna-COBL26*) or intron-retention (*BnaCOBL19* and *BnaCOBL24*). These splicing variants were confirmed in the reported rapeseed genomes [18]. Whether do structural change and alternative splicing cause neofunctionalization and/or subfunctionalization of AtCOBL4 subgroup members in rapeseed is worth more studies.

## Supporting information

**S1 Fig. Sequence alignment of the COBRA domain of COBL proteins in Arabidopsis, rice, corn and rapeseed.** *BC1_ rice_Japo* (AAQ56120.1) and *BC1_rice_Indi* (AAQ56121.1) represent the Brittle Culm1 protein in *Oryza sativa* subsp. Indica and *Oryza sativa* subsp. Japonica respectively. *BK2* (ABJ99754.1) encodes brittle_stalk-2 protein in corn.
(TIF)

**S2 Fig. Sequence alignment of AtCOB and AtCOBL4 orthologous subgroup proteins in Arabidopsis, B. rapa, B. oleracea and B. napus.** The blue square brackets contain the *COBL* gene across four *B. rapa* genomes; The green square brackets contain the *COBL* gene across three *B. oleracea* genomes; The purple-red square brackets contain the *COBL* gene across nine *B. napus* genomes. Conservative domains are in the rectangle. "*M1*", " *M2*", "*M3*" and " *M5*" are the crucial sites that were verified by mutants to *BC1* of rice; "*M4*" point at a transposon insertion site in BK2 of corn.
(SVG)

**S1 Table. Primer sequences designed for qRT-PCR of selected *BnaCOBLs.***
(XLSX)

**S2 Table. The basic information concerning COBLs in *B. napus*, *B. rapa*, and *B. oleracea*.**
(XLSX)

**S3 Table. Orthologous *COBL* gene pairs between *B. napus* with *B. rapa* and *B. oleracea*.**
(XLSX)

**S4 Table. *Cis*-acting regulatory elements related to stress, light and phytohormone responsiveness in the promoter region of *BnaCOBL* genes.**
(XLSX)

**S5 Table. TPM values of *BnaCOBLs* in different tissues and stems with distinct SBR.**
(XLSX)

## Author Contributions

**Conceptualization:** Hao Chen, Zhongsong Liu.

**Data curation:** Qian Yang.

**Formal analysis:** Qian Yang.

**Funding acquisition:** Hao Chen.

**Investigation:** Qian Yang.

**Methodology:** Hao Chen.

**Resources:** Zhongsong Liu.

**Software:** Shan Wang.

**Supervision:** Qian Yang.

**Validation:** Qian Yang, Liang You.

**Visualization:** Qian Yang.

**Writing – original draft:** Qian Yang.

**Writing – review & editing:** Fangying Liu, Zhongsong Liu.

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
