## [Decision Letter · Decision Letter 0]

21 May 2021

PONE-D-21-09114

Genome-wide identification of the COBRA-like gene family in rapeseed (Brassica napus L.)

PLOS ONE

Dear Dr. Liu,

Thank you for submitting your manuscript to PLOS ONE. After careful consideration, we feel that it has merit but does not fully meet PLOS ONE’s publication criteria as it currently stands. Therefore, we invite you to submit a revised version of the manuscript that addresses the points raised during the review process.

We look forward to receiving your revised manuscript.

Kind regards,

Kun Lu, Ph.D.

Academic Editor

PLOS ONE

Journal Requirements:

"This research was funded by the National Key Research and Development

Program of China (grant number: 2017YFD0101702), the Scientific Research

Foundation of the Hunan Provincial Education Department (grant number: 20A261)

and the Key Research and Development Program of Hunan Province (grant number:

2016JC2024)."

Reviewers' comments:

Reviewer's Responses to Questions

**Comments to the Author**

1. Is the manuscript technically sound, and do the data support the conclusions?

Reviewer #1: Partly

Reviewer #2: Yes

2. Has the statistical analysis been performed appropriately and rigorously? 

Reviewer #1: Yes

Reviewer #2: Yes

3. Have the authors made all data underlying the findings in their manuscript fully available?

Reviewer #1: Yes

Reviewer #2: Yes

4. Is the manuscript presented in an intelligible fashion and written in standard English?

Reviewer #1: Yes

Reviewer #2: Yes

5. Review Comments to the Author

Reviewer #1: In this manuscript, the authors conducted a genome-wide analysis of the COBRA-like gene family in rapeseed. They also performed phylogenetic analysis, cis-element, and gene expression analysis. However, I do have some concerns before its final acceptance.

1. In phylogenetic analysis COBLs, authors had done analysis using protein sequences from Brassica and A. thaliana only. They can draw a better picture/results if they use more species for phylogenetic analysis.

2. Why authors randomly selected the five genes out of 44 for Q-RT PCR analysis? In my opinion, there should be a criterion/logic for selection. Like, (a) selection based on their subclass division; phylogenetic based, (b) selection of genes based on their in-silico expression results.

3. “BnaCOBLs were involved in stem lodging resistance” based on these experiments, it overestimated results. In my opinion, some more experiments should be conducted to further functional validations of results to meet the journal standards. Functional validation of genes using transgenic approach/VIGS/CRISPAR.

4. It is also suggested that authors should improve the discussion section. They have to discuss their results and compare them with some earlier and recently published papers in more depth and clarity.

5. Please provide the Pfam ID used for COBL gene identification.

6. Detail of BnaActin gene used in this study. i.e., NCBI ID, or any other database ID should be provided.

7. How many biological replicates of each tissue were used for qRT-PCR analysis, since it is not clear from the martial and methods section. Please clear?

8. How they named COBL genes? Please provide details.

Reviewer #2: In this paper, using the method of biological information, 44 COBRA-like genes were identified in rapeseed. Then, the phylogenetic relationships, the chromosome location, the domain composition, and the putative cis element distribution of BnaCOBLs were determined. Different expression levels in stems with distinct lodging resistance suggested that there was some association between BnaCOBLs and stem lodging resistance. The logic of the article is very clear, and the scientific issues of concern are also well explained. But there are still several issues that need attention:

(1) In line 127, when determining the promoter, you should not only use the length as the standard, but should combine some basic characteristics of the promoter, such as the -10 region and the -35 region. Because the promoter may exist in the last gene, or the two genes mentioned in your article share the promoter. The sequence of the promoter directly affects the results of the subsequent analysis. For example, the speculation of gene function in line 445 may be biased.

(2) Line 117. Why you choosed 10 motifs and motif width of 8-50 amino-acid residues as this papaer's parameters. The basis should be pointed out here.

(3) In the discussion section, when the article directly quoted the content of the article results, the corresponding picture number was not marked. I think it will be easier for readers to read this article by marking the corresponding picture numbers.

6. PLOS authors have the option to publish the peer review history of their article (what does this mean?). If published, this will include your full peer review and any attached files.

Reviewer #1: No

Reviewer #2: **Yes: **Guanghui Xiao

---

## [Author Response · Author response to Decision Letter 0]

16 Aug 2021

Review Comments to the Author

Reviewer #1: In this manuscript, the authors conducted a genome-wide analysis of the COBRA-like gene family in rapeseed. They also performed phylogenetic analysis, cis-element, and gene expression analysis. However, I do have some concerns before its final acceptance.

1. In phylogenetic analysis COBLs, authors had done analysis using protein sequences from Brassica and A. thaliana only. They can draw a better picture/results if they use more species for phylogenetic analysis.

In this case, we revealed the evolutionary source and expansion of BnaCOBLs family mainly by the phylogenetic analysis, so we chose only A.thaliana and two diploid progenitor species B. rapa and B. oleracea. The COBL family had already discovered conserving in monocots and eudicots from other researches in rice, soybean and corn.

2. Why authors randomly selected the five genes out of 44 for Q-RT PCR analysis? In my opinion, there should be a criterion/logic for selection. Like, (a) selection based on their subclass division; phylogenetic based, (b) selection of genes based on their in-silico expression results.

We chose the five genes for Q-RT PCR analysis justly based on their phylogenetic subgroup division, and their in-silico expression. The BnaCOBL9, BnaCOBL35, and BnaCOBL41 belong to AtCOB subgroup; the BnaCOBL19 and BnaCOBL24 belong to AtCOBL4 subgroup. The members in both subgroups differently expressed in different levels of the stem breaking resistance. 

3. “BnaCOBLs were involved in stem lodging resistance” based on these experiments, it overestimated results. In my opinion, some more experiments should be conducted to further functional validations of results to meet the journal standards. Functional validation of genes using transgenic approach/VIGS/CRISPAR.

Thanks for your suggestion, we also thought the evidence was not sufficient. We changed the conclusion. 

4. It is also suggested that authors should improve the discussion section. They have to discuss their results and compare them with some earlier and recently published papers in more depth and clarity.

Yes, we have revised the discussion section in the manuscript. 

5. Please provide the Pfam ID used for COBL gene identification.

The Pfam ID of COBRA-like gene family is PF04833

6. Detail of BnaActin gene used in this study. i.e., NCBI ID, or any other database ID should be provided.

The expression levels of mRNAs were normalized to BnaActin (BnaA10g06670D), which has been listed on S1 Table.

7. How many biological replicates of each tissue were used for qRT-PCR analysis, since it is not clear from the martial and methods section. Please clear? 

We performed three biological replicates, which have been provided in line 158 of the manuscript.

8. How they named COBL genes? Please provide details. 

COBLs is the abbreviation of COBRA-like genes. The number suffixes were added according to the chromosome location from Chromosome A01 to Chromosome C09. This could also be shown on Figure 2.

Reviewer #2: In this paper, using the method of biological information, 44 COBRA-like genes were identified in rapeseed. Then, the phylogenetic relationships, the chromosome location, the domain composition, and the putative cis-element distribution of BnaCOBLs were determined. Different expression levels in stems with distinct lodging resistance suggested that there was some association between BnaCOBLs and stem lodging resistance. The logic of the article is very clear, and the scientific issues of concern are also well explained. But there are still several issues that need attention :

(1) In line 127, when determining the promoter, you should not only use the length as the standard, but should combine some basic characteristics of the promoter, such as the -10 region and the -35 region. Because the promoter may exist in the last gene, or the two genes mentioned in your article share the promoter. The sequence of the promoter directly affects the results of the subsequent analysis. For example, the speculation of gene function in line 445 may be biased.

After further reviewing the literature on promoter research, we identified cis-elements in all 1500bp upstream of the ATG codon and did not filter out the elements on the opposite strand. Since the cis-element may exist in the genes on the upstream. The elements in core promoter regions nearing the TSS (transcript start site) are direction-sensitive, but the elements in the distal region are direction-insensitive. 

We checked the TATA-box and CAAT-box in the proximal region (-500bp) of the start codon, only BnaCOBL10 and BnaCOBL36 loss their TATA-box. But the TATA-less type promoter takes account for 79.9% of all promters in Arabidopsis thaliana. So we temporarily identified the elements in the 1500bp upstream region of the start codon when batch analyzing. 

 (2) Line 117. Why you choosed 10 motifs and motif width of 8-50 amino-acid residues as this papaer's parameters. The basis should be pointed out here. 

Once, we had predicted the conserved motifs of COBL family by MEME software with these parameters. Then we use the known COBRA domains of COBL family to evaluate BnaCOBLs instead. This made the protein structural analysis more meaningful. 

(3) In the discussion section, when the article directly quoted the content of the article results, the corresponding picture number was not marked. I think it will be easier for readers to read this article by marking the corresponding picture numbers. 

Yes, we have marked these signs in the revised discussion section.

---

## [Decision Letter · Decision Letter 1]

26 Aug 2021

PONE-D-21-09114R1

Genome-wide identification and expression profiling of the *COBRA-like genes* reveal likely roles in stem strength in rapeseed (*Brassica napus*L.)

PLOS ONE

Dear Dr. Liu,

Thank you for submitting your manuscript to PLOS ONE. After careful consideration, we feel that it has merit but does not fully meet PLOS ONE’s publication criteria as it currently stands. Therefore, we invite you to submit a revised version of the manuscript that addresses the points raised during the review process.

We look forward to receiving your revised manuscript.

Kind regards,

Kun Lu, Ph.D.

Academic Editor

PLOS ONE

Journal Requirements:

Additional Editor Comments (if provided):

Reviewers' comments:

Reviewer's Responses to Questions

**Comments to the Author**

1. If the authors have adequately addressed your comments raised in a previous round of review and you feel that this manuscript is now acceptable for publication, you may indicate that here to bypass the “Comments to the Author” section, enter your conflict of interest statement in the “Confidential to Editor” section, and submit your "Accept" recommendation.

Reviewer #3: All comments have been addressed

Reviewer #4: All comments have been addressed

2. Is the manuscript technically sound, and do the data support the conclusions?

Reviewer #3: Yes

Reviewer #4: Yes

3. Has the statistical analysis been performed appropriately and rigorously? 

Reviewer #3: Yes

Reviewer #4: Yes

4. Have the authors made all data underlying the findings in their manuscript fully available?

Reviewer #3: Yes

Reviewer #4: Yes

5. Is the manuscript presented in an intelligible fashion and written in standard English?

Reviewer #3: Yes

Reviewer #4: Yes

6. Review Comments to the Author

Reviewer #3: Yang et al., used bioinformatics method to analyze the COBL gene family in rapeseed. The results are interesting, and may have potential guiding significance for lodging resistance improvement. The major comments are:

1. Whether the animal (like mouse or human) has COBL homeologues, if has, please introduce the gene function of COBL genes in Introduction.

2. The authors are better to give the uniform naming in B. napus, B. rapa and B. oleracea based on Fig. 1 result, and combine the Table S2 and S3 together. Otherwise the readers hard to compare the same homeologue in different species.

3. Based on the protein location predication result, the BnaCOBL10 and BnaCOBL42 were not on cell membrane, while these two genes were in the same subgroup (Fig. 2). The authors may discuss the possibility of the differentiation of gene function in Discussion.

4. The authors need to check the italics of the gene name fully (e.g. Line 37, 44, 56, 57, 79, 141, 223, 226, 229, 230 etc.)

Reviewer #4: In this manuscript, 44 COBRA-like genes were identified in rapeseed by sequence identity and conservative COBRA domain. The author characterized these genes on phylogenetic grouping, the chromosome location, the domains, and the putative cis-element distribution in promoters of BnaCOBLs. The expression profiles and the protein domains of AtCOB and AtCOBL4 subgroup imply these AtCOB subgroup members may be involved in stem development and stem breaking resistance of rapeseed, rather than members in AtCOBL4 subgroup, which are functional in crops of grass family. The logic of the article is clear, and the scientific issues of concern are also well explained. But I think authors should address following points in this manuscript before publishing in “ PLOS ONE ”:

1. The conclusion in line 79 to 80 seemed to be expanded according to the results and discusses.

2. As the topology of the phylogenetic tree (Fig 1), BnaCOBL5 and BnaCOBL44 did not belong to any subgroup orthologs from AtCOBLs. Where they evolved from? Maybe these two genes are not the members of COBL family in rapeseed.

3. The alternative splicing of subgroup AtCOBL4 presented in the Result section should be supported by figures.

4. Why the members in AtCOBL4 subgroup with incomplete protein still are expressed in stems at a certain level?

5. In line 19 in the Abstract section, whether the expansion of COBL family in B.napus was attributable to ‘whole-genome duplication’ or ‘whole-genome triplication’ as written in line 423?

6. Some syntax and formatting errors such as, in line 243, ‘The kind of gene pair was’ should revise as ‘This kind of gene pairs were’ in this context; in Table 1, the ‘p-value’ in line where the ‘Gene Name’ is ‘BnaCOBL20’, does not line up with other elements in the same line.

7. Add some correlation descriptions between the genes (such as BnaCOBL9, BnaCOBL35 and BnaCOBL24 etc.) expression change and stem strength in results.

7. PLOS authors have the option to publish the peer review history of their article (what does this mean?). If published, this will include your full peer review and any attached files.

Reviewer #3: No

Reviewer #4: No

---

## [Author Response · Author response to Decision Letter 1]

28 Sep 2021

Dear Referees:

We thank the reviewers for their enthusiasm and interest in this research. We would like to acknowledge the referees for spending their time and effort sharing their views and providing constructive comments. These comments help us improve our manuscript. The detailed response is provided below following the referee’s specific comments.

Reviewer #3: Yang et al., used bioinformatics method to analyze the COBL gene family in rapeseed. The results are interesting, and may have potential guiding significance for lodging resistance improvement. The major comments are:

1. Whether the animal (like mouse or human) has COBL homeologues, if has, please introduce the gene function of COBL genes in Introduction.

The COBRA-like gene family (pfam04833) is the only member of the superfamily cl04787 which only be found in embryophyte from a search of the conserved domain database (Search on 1st Sep, 2021). 

2. The authors are better to give the uniform naming in B. napus, B. rapa and B. oleracea based on Fig. 1 result, and combine the Table S2 and S3 together. Otherwise the readers hard to compare the same homeologue in different species.

As you recommended, we have renamed the COBL genes of B. rapa (20) and B. oleracea (23) as BraCOBL1-20 and BolCOBL1-23 based on their chromosome locations. We combined the Tables S2 and S3 together to form a new S2 Table. 

3. Based on the protein location predication result, the BnaCOBL10 and BnaCOBL42 were not on cell membrane, while these two genes were in the same subgroup (Fig. 2). The authors may discuss the possibility of the differentiation of gene function in Discussion.

Both BnaCOBL10 and BnaCOBL42 are orthologous from AtCOBL5, that is the only member without the GPIω-attachment site in Arabidpsis. The GPIω-attachment site is important for COBL proteins to be anchored in plasma membrane. So the orthologs of AtCOBL5 subgroup in rapeseed are not predicted to be on the plasma membrane. The function of the gene AtCOBL5 in Arabidpsis is little known, so did the orthologs of AtCOBL5 subgroup in other plants.

4. The authors need to check the italics of the gene name fully (e.g. Line 37, 44, 56, 57, 79, 141, 223, 226, 229, 230 etc.)

The words with capital letters in lines listed by reviewer, are not the names of genes but the names of subgroups or gene families. It is not necessary to italic the names of subgroups or gene families.

Reviewer #4: In this manuscript, 44 COBRA-like genes were identified in rapeseed by sequence identity and conservative COBRA domain. The author characterized these genes on phylogenetic grouping, the chromosome location, the domains, and the putative cis-element distribution in promoters of BnaCOBLs. The expression profiles and the protein domains of AtCOB and AtCOBL4 subgroup imply these AtCOB subgroup members may be involved in stem development and stem breaking resistance of rapeseed, rather than members in AtCOBL4 subgroup, which are functional in crops of grass family. The logic of the article is clear, and the scientific issues of concern are also well explained. But I think authors should address following points in this manuscript before publishing in “PLOS ONE ”:

1. The conclusion in line 79 to 80 seemed to be expanded according to the results and discusses.

Based on our results we redrawn the conclusion. 

2. As the topology of the phylogenetic tree (Fig 1), BnaCOBL5 and BnaCOBL44 did not belong to any subgroup orthologs from AtCOBLs. Where they evolved from? Maybe these two genes are not the members of COBL family in rapeseed.

The BnaCOBL5 and BnaCOBL44 contain the COBRA domain, which meets the criteria of a member of COBL family. In fact, analysis of identity indicated both genes evolve from AtCOBL2/3 orthologs. 

3. The alternative splicing of subgroup AtCOBL4 presented in the Result section should be supported by figures.

As shown in Fig4.tif, we illustrated the skipped exons in rectangle with dotted line. 

4. Why the members in AtCOBL4 subgroup with incomplete protein still are expressed in stems at a certain level?

This phenomenon is caused by the unchanged promoters of members in AtCOBL4 subgroup. 

5. In line 19 in the Abstract section, whether the expansion of COBL family in B. napus was attributable to ‘whole-genome duplication’ or ‘whole-genome triplication’ as written in line 423?

It is ‘whole-genome triplication’ in line 19.

6. Some syntax and formatting errors such as, in line 243, ‘The kind of gene pair was’ should revise as ‘This kind of gene pairs were’ in this context; in Table 1, the ‘p-value’ in line where the ‘Gene Name’ is ‘BnaCOBL20’, does not line up with other elements in the same line.

We carefully checked and corrected the grammatical and formatting errors in the revised manuscript.

7. Add some correlation descriptions between the genes (such as BnaCOBL9, BnaCOBL35 and BnaCOBL24 etc.) expression change and stem strength in results.

We have described the expression change patterns in line 376 to 382.

---

## [Decision Letter · Decision Letter 2]

8 Nov 2021

Genome-wide identification and expression profiling of the *COBRA-like * genes reveal likely roles in stem strength in rapeseed (*Brassica napus* L.)

PONE-D-21-09114R2

Dear Dr. Liu,

We’re pleased to inform you that your manuscript has been judged scientifically suitable for publication and will be formally accepted for publication once it meets all outstanding technical requirements.

Kind regards,

Kun Lu, Ph.D.

Academic Editor

PLOS ONE

Additional Editor Comments (optional):

Reviewers' comments:

Reviewer's Responses to Questions

**Comments to the Author**

1. If the authors have adequately addressed your comments raised in a previous round of review and you feel that this manuscript is now acceptable for publication, you may indicate that here to bypass the “Comments to the Author” section, enter your conflict of interest statement in the “Confidential to Editor” section, and submit your "Accept" recommendation.

Reviewer #3: All comments have been addressed

Reviewer #4: All comments have been addressed

2. Is the manuscript technically sound, and do the data support the conclusions?

Reviewer #3: Yes

Reviewer #4: Yes

3. Has the statistical analysis been performed appropriately and rigorously? 

Reviewer #3: Yes

Reviewer #4: Yes

4. Have the authors made all data underlying the findings in their manuscript fully available?

Reviewer #3: Yes

Reviewer #4: Yes

5. Is the manuscript presented in an intelligible fashion and written in standard English?

Reviewer #3: Yes

Reviewer #4: Yes

6. Review Comments to the Author

Reviewer #3: Yang et al. addressed all the comments, and revised the manuscript fully. The manuscript is matched the publication criteria of PLOS ONE right now.

Reviewer #4: This paper is focused on identification and expression profiling of the COBRA-like  genes reveal likely roles in stem strength in rapeseed, it is well organized and its presentation is good. I think it meets the current publication criteria of PLOS ONE.

7. PLOS authors have the option to publish the peer review history of their article (what does this mean?). If published, this will include your full peer review and any attached files.

Reviewer #3: No

Reviewer #4: No

---

## [Editor Report · Acceptance letter]

16 Nov 2021

PONE-D-21-09114R2 

Genome-wide identification and expression profiling of the *COBRA-like * genes reveal likely roles in stem strength in rapeseed (*Brassica napus* L.) 

Dear Dr. Liu:

I'm pleased to inform you that your manuscript has been deemed suitable for publication in PLOS ONE. Congratulations! Your manuscript is now with our production department. 

Kind regards, 

on behalf of

Dr. Kun Lu 

Academic Editor

PLOS ONE